# Future-proof vaccine design with a generative model of antibody cross-reactivity

Noor Youssef [* 1 2]   Sarah Gurev [* 1 2 3]   Hannah Pierce-Hoffman [4]   Alexander A. Cohen [5]   Luis F. Caldera [5]
Pamela J. Bjorkman [5]   Debora S. Marks [1 2]

## Abstract

Mosaic nanoparticle vaccines incorporating naturally diverse sarbecovirus receptor binding domains (RBDs) represent a promising approach for pan-coronavirus vaccines. Mosaic nanoparticles elicit broad, cross-reactive immune responses, likely because elicited antibodies utilize avidity effects to preferentially bind conserved regions across neighboring RBDs. However, the diversity in natural RBDs is limited, leading to 'off-target' antibodies that bind to conserved regions across the selected RBDs but which are likely to mutate in the future. We therefore develop a novel future-proof vaccine design method, building upon a probabilistic generative model of antibody escape, to computationally design RBDs with further diversity. This approach aims to focus antibody responses to regions that are (1) neutralizing, (2) accessible to antibodies during a natural infection and (3) unlikely to mutate during future viral evolution. The designs will be assessed by immunizing mice and testing the breadth of neutralizability of the sera compared to a nanoparticle composed of naturally diverse strains.

## 1. Introduction

The COVID-19 pandemic, particularly the waning immunity from SARS-CoV-2 vaccines, reinforces the need for universal vaccines that are broadly protective against diverse members of a viral family. A pan-coronavirus vaccine would ideally not only protect against presently circulating SARS-CoV-2 variants but also against future variants and other species within the family. This includes existing endemic human coronaviruses and potential spill-over events that could seed a new pandemic. This urgent need is underscored by the billions of dollars earmarked for the development of pan-coronavirus vaccines across a range of approaches including mRNA-based, nanoparticle-based, and protein subunit vaccines (Dolgin, 2022; Musunuri et al., 2024; Caradonna & Schmidt, 2021).

Mosaic nanoparticles represent one such promising approach for pan-coronavirus vaccines, as well as other pan-genus or pan-family vaccines for viruses such as influenza. In a nanoparticle-based vaccine, antigenic proteins are attached to a central protein scaffold (Brune & Howarth, 2018). This vaccine modality inherently lends itself to broad protection in two crucial ways. Firstly, in the case of a mosaic nanoparticle, different antigens are presented, from diverse viral species (as opposed to a homotypic nanoparticle where each antigen is the same). Secondly, when adjacent antigens differ, antibodies capable of cross-linking between conserved regions in neighboring proteins, using avidity binding, are elicited (Fig 1A). These conserved epitopes, shared across viral species, are hypothesized to be less prone to future mutation and antibodies binding them are more likely to protect against the entire viral family, even beyond strains present on the nanoparticle (Cohen et al., 2021a; 2022). Nanoparticles containing 8 naturally diverse sarbecovirus RBDs elicit broader cross-reactive responses compared to homotypic nanoparticles or convalescent plasma. This broader response offers protection both against other sarbecoviruses and SARS-CoV-2 variants not present on the nanoparticle (Cohen et al., 2021a; 2022; 2024).

Despite its potential, the current mosaic nanoparticle presents significant challenges. First, there is evidence of waning immunity to newer SARS-CoV-2 variants (Cohen et al., 2024). Second, the nanoparticle elicits off-target antibodies, many of which are inaccessible in the context of the full Spike protein encountered during an infection, failing to provide effective protection. Additionally, epitopes may be conserved between a pair of viruses on the nanoparticle despite a lack of conservation throughout the sarbecovirus sub-genus, highlighting potential susceptibility to future mutations (Cohen et al., 2021a; 2022).

---

[*]Equal contribution [1]Systems Biology, Harvard Medical School, Boston, Massachusetts, USA [2]Broad Institute of Harvard and MIT, Cambridge, Massachusetts, USA [3]EECS, Massachusetts Institute of Technology, Cambridge, Massachusetts, USA [4]Biomedical Informatics, Harvard Medical School, Boston, Massachusetts, USA [5]Biology and Biological Engineering, Caltech, Pasadena, California, USA. Correspondence to: Noor Youssef <noor youssef@hms.harvard.edu>, Sarah Gurev <sgurev@mit.edu>, Debora S. Marks <debbie@hms.harvard.edu>.

*Accepted at the 1st Machine Learning for Life and Material Sciences Workshop at ICML 2024.* Copyright 2024 by the author(s).

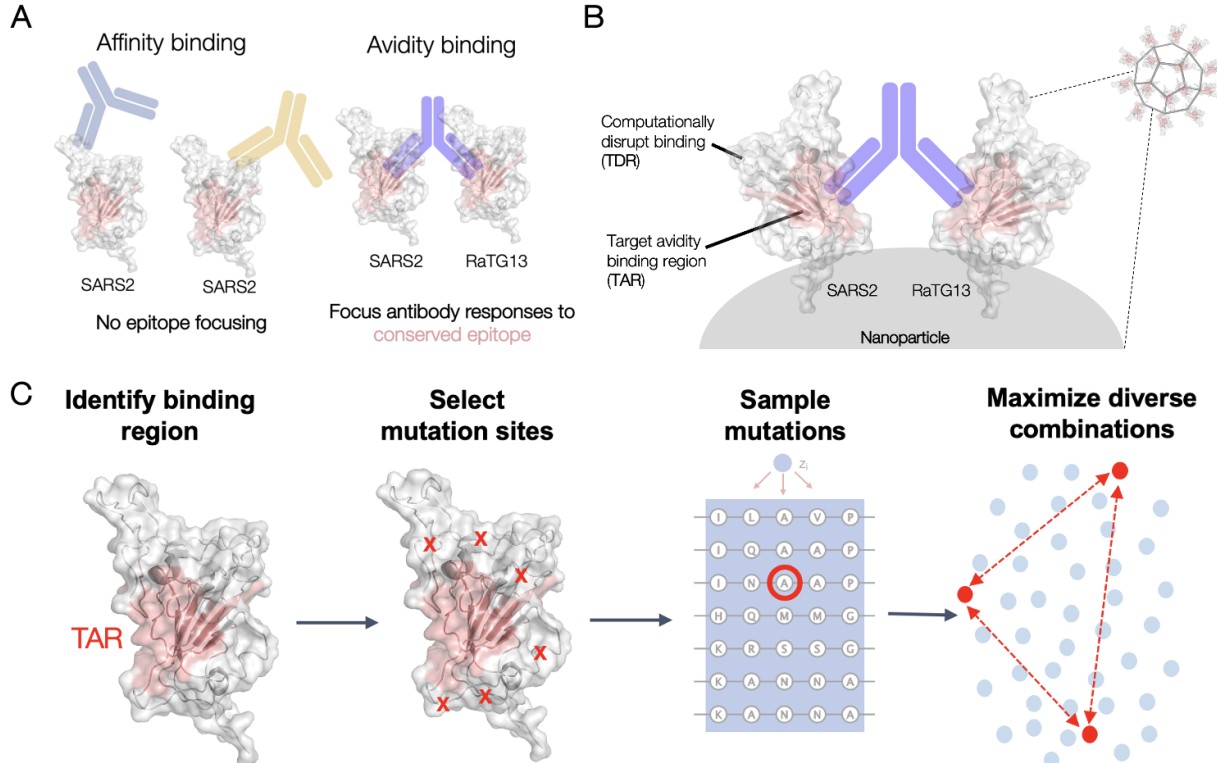

*Figure 1.* **A**. A single RBD elicits antibodies that bind with affinity to many regions. Conversely, two adjacent, different RBDs focuses antibody responses to conserved regions across the RBDs through avidity effects. **B**. RBD antigens were designed to have a Target Avidity Region (TAR) that is conserved, neutralizing, and less likely to mutate during viral evolution. To optimize antibody binding to the TAR we computationally designed mutations elsewhere (in the Target Disruption Region or TDR) to prevent off-target binding. **C**. Steps in nanoparticle design process.

We propose to address these challenges by designing mosaic RBD nanoparticles that focus the antibody response. Our approach involves generating RBDs with greater diversity than natural strains outside a selected conserved, neutralizing region. This is aimed at refining antibody responses to focus on regions that are accessible and less likely to mutate, focusing antibodies beyond what is naturally conserved between any pair of natural strains on the original nanoparticle. Prior approaches to masking undesirable epitopes (Caradonna & Schmidt, 2021), including hyperglycosylation (Thornlow et al., 2021) and random mutation with relatively inert amino acids (Frei et al., 2018), have achieved moderate success for other vaccine modalities.

Given the substantial number of mutations to optimize in a mosaic nanoparticle, we build upon our prior work creating a deep generative model of antibody escape, EVEscape (Thadani et al., 2023; Youssef et al., 2023), to tackle this novel vaccine design approach computationally (Fig 1B). EVEscape, trained on pre-pandemic data, has been shown to be as accurate as high-throughput experimental scans using pandemic antibodies at anticipating SARS-CoV-2 mutations escaping prior immunity (Thadani et al., 2023).

EVEscape has also demonstrated remarkable success in designing antigenically diverse SARS-CoV-2 sequences for vaccine testing which remain infectious at distances of up to 46 mutations from the ancestral strain (Youssef et al., 2023).

While EVEscape was framed previously as a model of antibody escape from prior infection or vaccination, it can more generally be considered to be a measure of cross-reactivity between antigens–whether antibodies directed against a first antigen can successfully bind a different antigen. In the case of modeling antibody escape in a pandemic, the first strain is the ancestral strain and the second is a future variant. However, it can be applied to cross-reactivity between two antigens on a single nanoparticle. Therefore, EVEscape provides an ideal foundation for an approach focused on generating functional heterotypic antigens with targeted mutations that eliminate cross-reactivity to any potential off-target epitope.

## 2. Design methodology

Here we address a novel sequence design problem: given a set of naturally diverse antigens in a viral family, identify

a region to target antibody binding and generate mutations elsewhere to minimize off-target antibodies, while retaining antigen functionality. The mutations should minimize the pairwise antibody cross-reactivity (where antibodies can bind with avidity) at any possible off-target epitope. The goal is to design a set of RBDs that, when immunizing as a mosaic nanoparticle, generates sera that is more broadly neutralizing against diverse existing and potential future species in a viral family, compared to a mosaic nanoparticle composed of the original, natural strains. We base our designs on the natural mosaic sarbecoviruses used previously: SARS-CoV-2 Beta, RaTG13, Rs4081, SHC014, Pang17, RmYN02, Rf1, and WIV1 (Cohen et al., 2021a; 2022; 2024). Critically, the model is unsupervised and does not rely on prior knowledge of specific epitopes or experimental data.

We develop a multi-step design process (Fig 1C)–rather than a joint optimization that directly results in a final group of designed antigens–due to the complexity of the design objectives. Moreover, with diverse antigens as the key goal, it is important to down-sample from many potential designs to avoid prematurely constraining design choices based on initially selected sequences. This design process builds upon EVEscape, which models antibody escape as the product of three components: (1) mutation maintains fitness, (2) mutation is accessible to antibodies, (3) mutation is sufficiently dissimilar to disrupt antibody binding (Thadani et al., 2023). By virtue of EVEscape's flexible, modular framework, we repurpose each of its components, separately and in combination, in the relevant steps of our approach.

### 2.1. Identify target avidity binding region

We first identify an optimal antibody binding region as a contiguous, accessible region with (1) physicochemical conservation across sarbecoviruses and (2) neutralization potential. Sequence conservation is often measured via Shannon entropy, calculated from the relative frequency of each amino acid in a multiple sequence alignment position. However this conservation ignores physicochemical similarities between some amino acids. We therefore develop a physicochemical-based conservation using the dissimilarity component from EVEscape (Thadani et al., 2023) to calculate a von Neumann Entropy (VNE) (Caffrey et al., 2004). This considers not only the relative frequencies $f$ of each amino acid $i$, but also a physicochemical amino acid similarity matrix $P$, such that

$$VNE = \sum_i \lambda_i \log_{20} \lambda_i,$$

where $\lambda_i$ are the eigenvalues of $\mathrm{diag}(f)P$.

Second, since most neutralizing antibodies for the RBD physically block the host receptor ACE2 from binding, we use distance to the ACE2 binding site as a proxy for neutralization potential. This definition of neutralizing potential is specific to the sarbecovirus RBDs, but can be generalized to consider minimum distance to other receptors or known neutralizing hotspots generally (e.g., from analyzing experimental antibody data).

Amongst accessible, non-Spike contacting positions we create a contiguous region (residues neighboring in the structure rather than necessarily in the sequence) by selecting patches (all residues within a radius of 8Å) with a high rank product of neutralization and conservation scores. This resulted in a target avidity binding region (TAR) of 66 residues, approximately one third of the RBD. This selected TAR has several advantages over the regions in each pair of natural sequences where antibodies bind with avidity: the TAR is composed of more non-Spike contacting positions (residues that are accessible even in the context of the full Spike trimer), contains residues that are not just similar across a given pair of natural sequences on the nanoparticle but have physicochemical conservation across the entire sarbecovirus family (with sequences identity as low as 68%), and ignores decoy epitopes from the natural strains that may be conserved but are of low neutralizing potential.

### 2.2. Select mutation sites

In order to minimize antibody cross-reactivity to pairs of antigens outside the TAR, we select combinations of positions to optimally eliminate off-target binding, which we call the target disruption region (TDR). We begin by identifying a region that due to great evolutionary diversity, is already sufficiently diverse to mitigate off-target binding, reducing the number of added mutations. We call this the evolutionary target disruption region (eTDR). We then select positions to mutate in the remaining region that is more likely to have off-target binding, which we call the designed TDR or dTDR.

First, we identify the already sufficiently dissimilar evolutionary TDR through the same patch-based approach described above, but from physicochemical-conservation of the natural strains on the nanoparticle alone. This creates an eTDR of 66 residues, with 87 residues remaining in neither the eTDR or TAR. Second, we identify positions to mutate within the designed TDR. In order to disrupt each potential epitope, we wish to select spatially dispersed sites to mutate. To do so, we use k-means clustering on the Cartesian coordinates of the alpha carbon of each residue – such that each cluster represents a potential off-target epitope to disrupt.

We find that using 8 centroids generates clusters in which the maximum distance between residues is less than ∼25Å, the typical diameter of conformational B cell epitopes (Sun et al., 2011). Within each of the 8 spatially distributed clusters, we select one position to mutate by centroid proximity.

We rerun the k-means algorithm with 5000 random initializations to generate many sets of mutation sites to use to create designs. Each of the generated designs has 8 spatially distributed mutations.

## 2.3. Sample mutations

After selecting sites to mutate, we create designed sequences by making single amino acid substitutions at the pre-selected sites using a probabilistic generative model, EVE (Frazer et al., 2021). EVE is a deep variational autoencoder trained on a multiple sequence alignment of evolutionarily related protein sequences, in this case sarbecovirus RBD sequences aligned to each of the base natural sequence antigens, which learns the constraints that underpin structure and function. We explore the impact of different fitness models in Fig S6.

At each pre-selected mutation site $h$, we use the EVE model to calculate the fitness effect of all possible amino acid substitution.

$$p^h = < p_1^h, ..., p_{20}^h >$$

The fitness of a protein sequence is quantified using the log likelihood ratio of the mutated sequence over that of the wildtype sequence. We then use three sampling strategies (fittest, random, and sampled) for generating designs to maximize diversity and full-sequence fitness. The first strategy involves creating designs where at each pre-selected site the fittest possible single mutation was chosen ($\mathbb{S}_{\text{fittest}}$). The second strategy involves selecting mutations at random (assigning equal probabilities to all mutations) at each mutational site without considering fitness ($\mathbb{S}_{\text{random}}$). Finally, for the third strategy, we perform a multinomial draw with substitution probabilities proportional to their fitness effect ($\mathbb{S}_{\text{sampled}}$), only considering mutations with scores above the model's median single mutation score. We remove any designs where no mutation in a cluster is fit. Finally, we down-sample our designs generated with the three strategies to the set with full-sequence EVE score (moving beyond single mutations to scores of 8 mutations in combination) higher than the median score of $\mathbb{S}_{\text{fittest}}$ and maximum score of $\mathbb{S}_{\text{random}}$.

## 2.4. Optimize combinations of designs

The final step is selecting groups of diverse sequences to place on the nanoparticle together from the large set of sampled designs. We wish to choose pairs of design sequences $s_j, s_k$ that maximize $d[s_j, s_k]$, a joint accessibility and physicochemical dissimilarity product across all off-target epitopes $\mathbb{E}$:

$$d[s_j, s_k] = \sum_{e \in \mathbb{E}} \text{acc}[e] \cdot \text{dist}[e, s_j, s_k]$$

$$\text{acc}[e] = \frac{1}{|e|} \sum_{r \in e} a[r]$$

$$\text{dist}[e, s_j, s_k] = \frac{1}{|e|} \sum_{r \in e} a[r] \cdot P'[s_j[r], s_k[r]]$$

where $\text{acc}[e]$ is the accessibility of epitope $e$, $\text{dist}[e, s_j, s_k]$ is the epitope difference between $s_j$ and $s_k$, $a[r]$ is the accessibility of residue $r$, $P'[s_j[r], s_k[r]]$ is the physicochemical dissimilarity between amino acids in position $r$, and $|e|$ is the number of residues in the epitope. Each potential off-target epitope corresponds to a patch around each residue in the target disruption region–where a patch includes all residues within $\sim 10$Å. Each epitope difference is weighted by the average accessibility of that epitope, as a measure of the relative importance of each epitope. We use the accessibility and physicochemical dissimilarity components from EVEscape (Thadani et al., 2023), having already selected putatively functional sequences using EVEscape's generative fitness model in the prior step.

To select a mosaic group of designs, we therefore create an accessibility-dissimilarity distance matrix between all pairs of designs. We then use a greedy algorithm to find a solution to the maximum dispersion problem, which seeks to maximize the sum of pairwise distances between selected designs. We start by choosing a random sequence $s$ to add to the mosaic nanoparticle $\mathbb{N}$. We continue to add to $\mathbb{N}$ by iteratively selecting the next design which maximizes the minimum distance to all designs already in $\mathbb{N}$ (that uses a different target virus from those already chosen). We repeated this process with different starting sequences and ultimately selected 10 groups of designs $\mathbb{N}$ for experimental characterization for their neutralization potential. Fig S1 showcases a representative group of nanoparticle designs (diverse, natural sarbecoviruses each with 8 different mutations) mapped onto a single RBD structure to see the different mutations sites across viruses.

## 3. Results

The ultimate evaluation of this approach will be producing nanoparticles including our designed RBDs, immunizing mice with the designed nanoparticles, and subsequently evaluating neutralization breadth from elicited antibodies. Such preclinical evaluations can be both time and resource intensive. In the meantime, we have begun with several computational evaluations and experimental evaluation of expression. We require each antigen to express and fold into a relatively proper conformation (note that the RBDs do not need to be functional enough to be infective or bind ACE2). Additionally, by spatially dispersing the mutations and mutating an accessible surface (rather than in the core

of a protein), we make it more likely for the protein to fold, as clusters of nearby mutations are more likely to experience negative epistasis and buried mutations are under more structural constrains.

To determine the most suitable model for sampling mutations, we tested six models: position-specific scoring matrix (PSSM), EVCouplings (Hopf et al., 2017), TranceptEVE (Notin et al., 2022), EVE (Frazer et al., 2021), Progen2 (Nijkamp et al., 2023) and ESM-1v (Meier et al., 2021). PSSM, EVCouplings (Hopf et al., 2017), and EVE (Frazer et al., 2021) are alignment-based models which consider each position in isolation, include pairwise interactions, or higher-order interactions, respectively. Progen2 (Nijkamp et al., 2023) and ESM-1v (Meier et al., 2021) are two state-of-the-art transformer-based protein language models which use natural language processing techniques to predict the likelihood of observing a given protein sequence given the lexicon of existing protein sequences. However, viral sequences are relatively underrepresented in the protein sequence universe, especially as only sequences from Uniref90 (which does not consider separately sequences that are more than 90% identical, as is true for much of viral variation) are included in training. Finally, we include TranceptEVE (Notin et al., 2022), a hybrid model which combines language modeling with alignment-based modeling. We evaluated the performance of the six models against expression from deep mutational scans for SARS-CoV-1 and SARS-CoV-2 (Starr et al., 2020; Starr, 2024) (Table 1). The best performing models were EVE and TranceptEVE for both viruses. Due to its interpretability, and our inclusion of only substitution mutations, we use EVE for our construct design.

*Table 1.* Spearman correlations between mutation effect predictions using different fitness models and experimental values from RBD expression deep mutational scanning experiments for SARS-CoV-1 (Starr, 2024) and SARS-CoV-2 (Starr et al., 2020). Corresponding scatter plots in Figs S2 and S3.

| Model | SARS-CoV-2 | SARS-CoV-1 |
|---|---|---|
| ESM-1v | -0.01 | -0.09 |
| Progen2 | 0.36 | 0.48 |
| PSSM | 0.37 | 0.36 |
| EVCouplings | 0.45 | 0.43 |
| EVE | 0.51 | 0.49 |
| TranceptEVE | 0.52 | 0.50 |

We first verify that selected mutations in designs are above a 50% threshold of all single mutations using the EVE fitness model for each target virus (Fig S4). The fitness score of the full RBD for each design is also higher than all generated antigens with random mutations at selected sites (Figs 2, S5). We explore the impact of different fitness models on designs (Fig S6). We also analyze the spatial

separation of the mutations and calculated joint accessibility and physicochemical distances, in order to maximize the impact on each potential off-target epitope (Figs S1, S7).

We have so far experimentally tested 3 of 10 nanoparticle sets of designs for expression. Approximately 22% (6/27) of the tested constructs were successfully expressed, with five of the nine viruses (SARS-CoV-1, SHC014, RmYN02, Rf1, and RaTG13) having at least one successful design (each with 8 mutations). This is especially remarkable considering the lack of known sequence diversity near these sarbecoviruses, most with less than five known strains within 10 mutations (Fig S8).

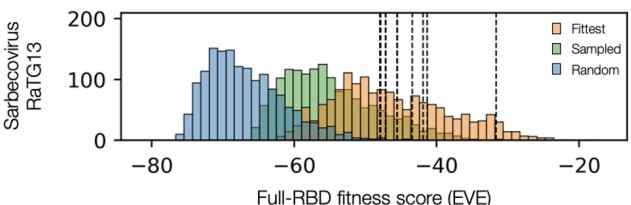

*Figure 2.* For each of the 8 natural RBDs in the mosaic-8b nanoparticle vaccine, we generate 15k sequences (5k by random amino acids, 5k by fittest amino acid, 5k by sampling proportional to fitness). Full-RBD fitness scores for 10 selected designs (dashed lines) on the RaTG13 sequence, relative to distributions $\mathbb{S}_{random}$, $\mathbb{S}_{fittest}$, and $\mathbb{S}_{sampled}$. Note that some lines are overlapping. Distributions for remaining RBDs in Fig S5.

## 4. Discussion

We present a novel future-proof vaccine design method using a deep generative model of antibody escape, EVEscape. Generative models are often divorced from real world applications. By wrapping our generative model into a larger, multi-step optimization process, which we then use to design a mosaic nanoparticle vaccine for experimental testing, we bridge this gap. Given EVEscape's modular components, we are able to build our design process on its foundation.

We will perform experimental testing by immunizing mice with our designed mosaic nanoparticle, evaluating our vaccine based on its neutralization breadth, against both diverse current and future strains. While we focus on the sarbecovirus sub-genus here, we will later explore our ability to achieve broader protection–for instance of the betacoranvirus genus or indeed of the entire coronavirus family. We will evaluate the breadth of protection relative to a nanoparticle composed of natural strains. To facilitate this comparison, we began our optimization from the same set of natural strains as in the vaccine developed by Cohen et al. (2021a; 2022). Once the success of our approach has been affirmed, an additional step in the design process could be

incorporated to select an initial set of diverse viral species as the base of the designs. Moreover, given our initial experimental results, we plan to iterate on our entire design process.

Thus far, our focus has been only on protection by neutralizing antibodies (both in our design objective and evaluation criteria), rather than protection from other sources, such as non-neutralizing antibody (e.g., ADCC) or T cell responses. Current vaccine practices take this same narrow focus–likely to their detriment. This may be beginning to change, especially for the pursuit of universal vaccines. By virtue of our multi-step process, it is simple for us to likewise incorporate additional design objectives, which we would then match with corresponding evaluation criteria. For example, our current design protocol does not assess the likelihood of triggering an autoimmune response against human proteins. A deimmunization assessment can be easily incorporated into our design protocol to ensure that none of the designed RBDs resemble human proteins.

Remarkably, the field of vaccine design has not yet utilized machine learning efforts to model antibody escape and cross-reactivity. Our approach is readily generalizable – both to other viral families (e.g., influenza nanoparticles already exist and could be similarity optimized (Cohen et al., 2021b)) and to other approaches for choosing nanoparticle antigens (including the full Spike, sub-dominant regions, or even other antigenic proteins like nucleocapsid). Furthermore, this process of optimizing nanoparticles or comparable heteromultimeric vaccination strategies (Lamson et al., 2023) are just the beginning of methods in which EVEscape or other generative models could benefit vaccine design. This is not even considering the new paradigm in generative models for vaccine testing efforts as outlined by Youssef et al. (2023). Indeed, we will also use EVEscape to design future strains to test protection, beyond evaluating against existing coronavirus diversity, as was previously the only option. The exciting possibilities presented by generative models, including EVEscape, hold immense potential for transforming vaccine design practices, promising far-reaching impacts across applications.

## Acknowledgements

The authors thank Nicole Thadani and other members of the Marks lab. This work was supported by the Coalition for Epidemic Preparedness Innovations (CEPI).

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
