# Supplementary Material
# Future-proof vaccine design with a generative model of antibody cross-reactivity

**Noor Youssef** [* 1 2] **Sarah Gurev** [* 1 2 3] **Hannah Pierce-Hoffman** [4] **Alexander A. Cohen** [5] **Luis F. Caldera** [5]
**Pamela J. Bjorkman** [5] **Debora S. Marks** [1 2]

## Glossary

| | | |
|---|---|---|
| ACE2 | Angiotensin-converting enzyme 2 | Human host receptor for SARS-CoV-2 entry |
| dTDR | Design TDR | Region to add mutations to disrupt off-target antibody avidity binding |
| eTDR | Evolutionary TDR | Region with sufficient evolutionary diversity to already disrupt off-target antibody avidity binding |
| RBD | Receptor Binding Domain | Domain of SARS-CoV-2 and other sarbecovirus Spike proteins used for nanoparticle vaccine - an immunodominant, neutralizing region |
| $\mathbb{S}_{fittest}$ | Fittest sequences | Sequence set with fittest mutation at chosen sites |
| $\mathbb{S}_{random}$ | Random sequences | Sequence set with random mutation at chosen sites |
| $\mathbb{S}_{sampled}$ | Sampled sequences | Sequence set with mutations sampled by fitness scores at chosen sites |
| TAR | Target Avidity Region | Region to target antibody avidity binding towards, where antibodies are likely to be protective across current and future sarbecoviruses |
| TDR | Target Disruption Region | Region to disrupt off-target antibody avidity binding - where high mutability, inaccessibility, or lack of neutralization potential would lead to less protective antibodies in the future |
| VNE | von Neumann Entropy | Physicochemical-based amino acid conservation |

[*]Equal contribution [1]Systems Biology, Harvard Medical School, Boston, Massachusetts, USA [2]Broad Institute of Harvard and MIT, Cambridge, Massachusetts, USA [3]EECS, Massachusetts Institute of Technology, Cambridge, Massachusetts, USA [4]Biomedical Informatics, Harvard Medical School, Boston, Massachusetts, USA [5]Biology and Biological Engineering, Caltech, Pasadena, California, USA. Correspondence to: Noor Youssef <noor youssef@hms.harvard.edu>, Sarah Gurev <sgurev@mit.edu>, Debora S. Marks <debbie@hms.harvard.edu>.

*Accepted at the 1st Machine Learning for Life and Material Sciences Workshop at ICML 2024.* Copyright 2024 by the author(s).

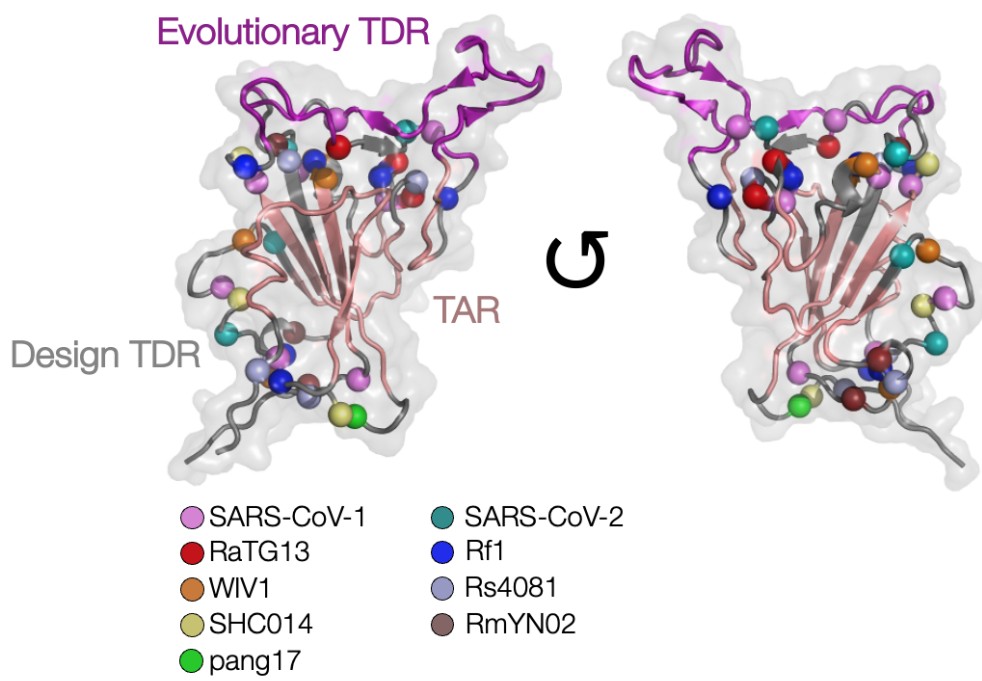

*Figure 1.* Representative group of nanoparticle designs (diverse, natural sarbecoviruses each with 8 different mutations) mapped onto a single RBD structure to see the different mutations sites across viruses. Within target viruses mutations are spatially distributed, and different sets of mutation sites are chosen for different target viruses. Note, a single color is shown even for overlapping positions.

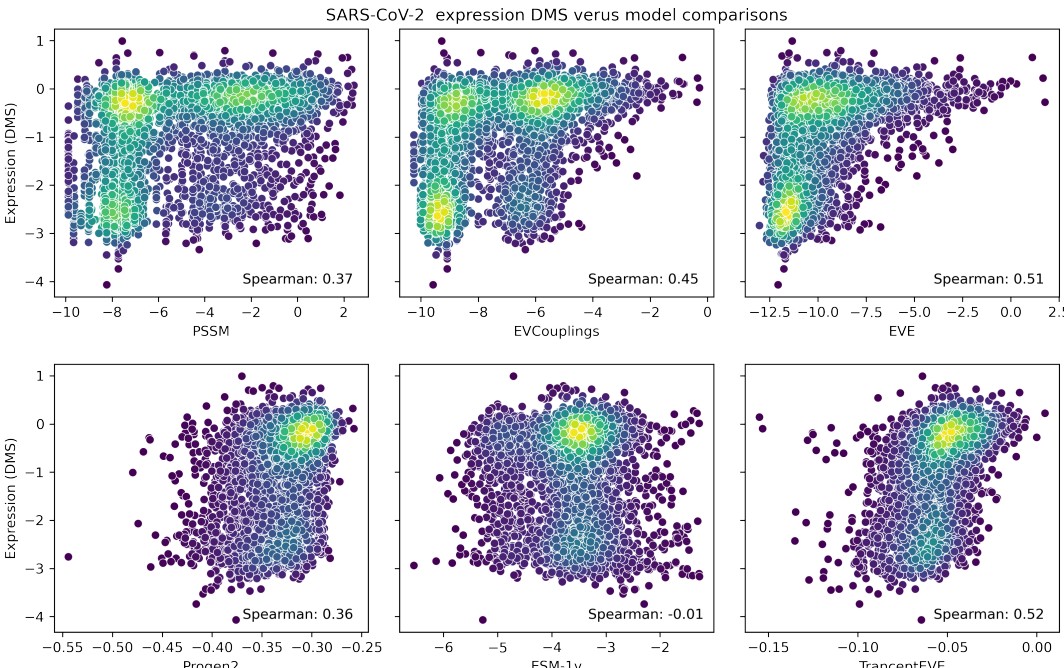

Figure 2. Correlations between fitness models and SARS-CoV-2 RBD deep mutational scans for expression (Starr et al., 2020).

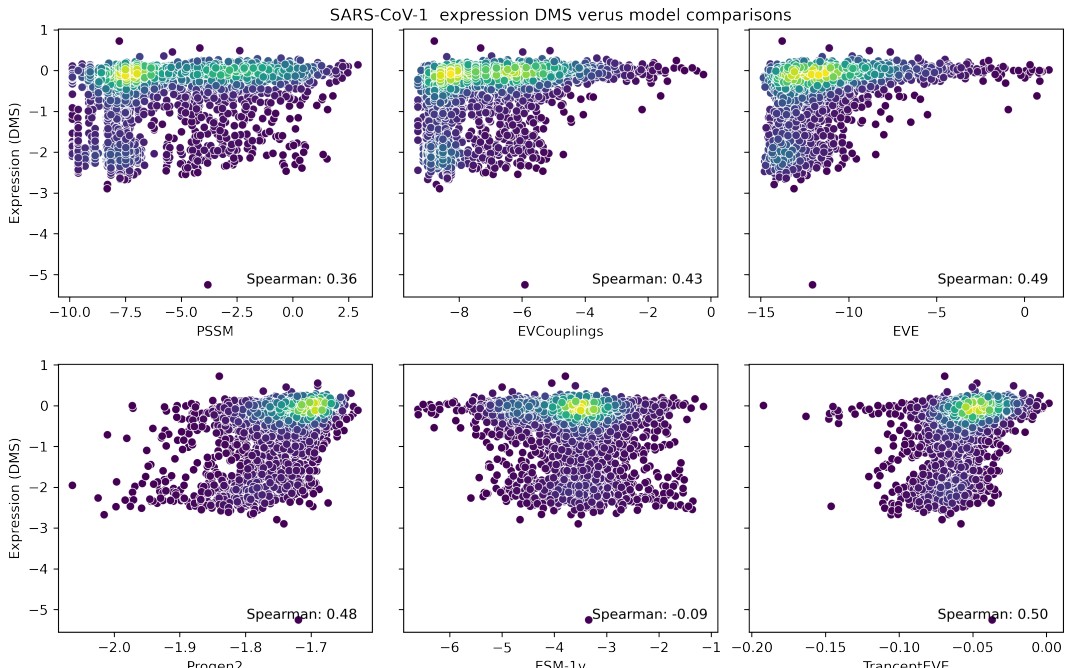

*Figure 3.* Correlations between fitness models and SARS-CoV-1 RBD deep mutational scans for expression (Starr, 2024).

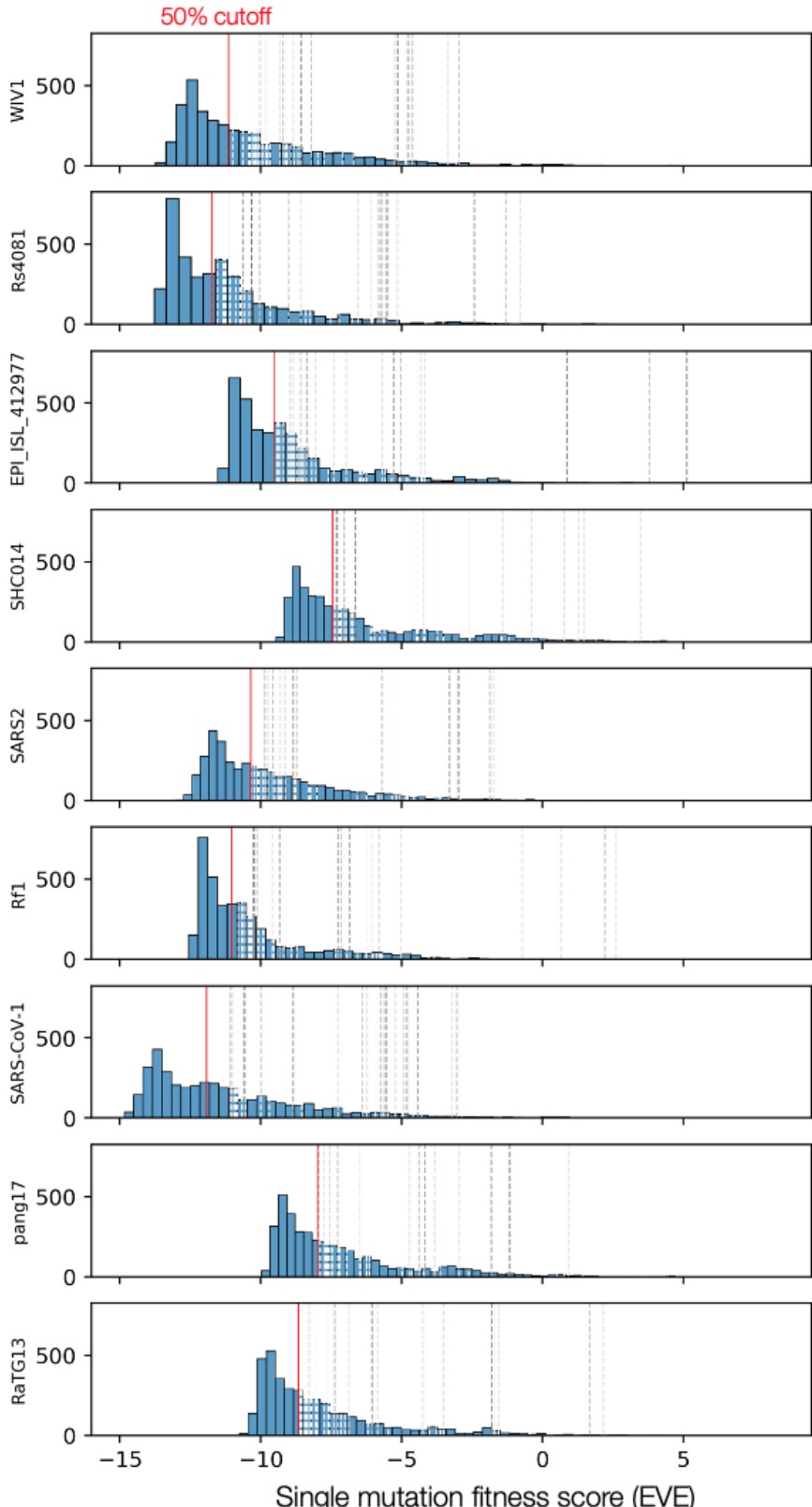

*Figure 4.* Distribution of single mutation fitness scores with EVE model (Red line indicates median score). Dashed lines show scores of selected mutations across 10 selected nanoparticle groups across the different target viruses (darker lines indicate a mutation that is in more than one selected nanoparticle group).

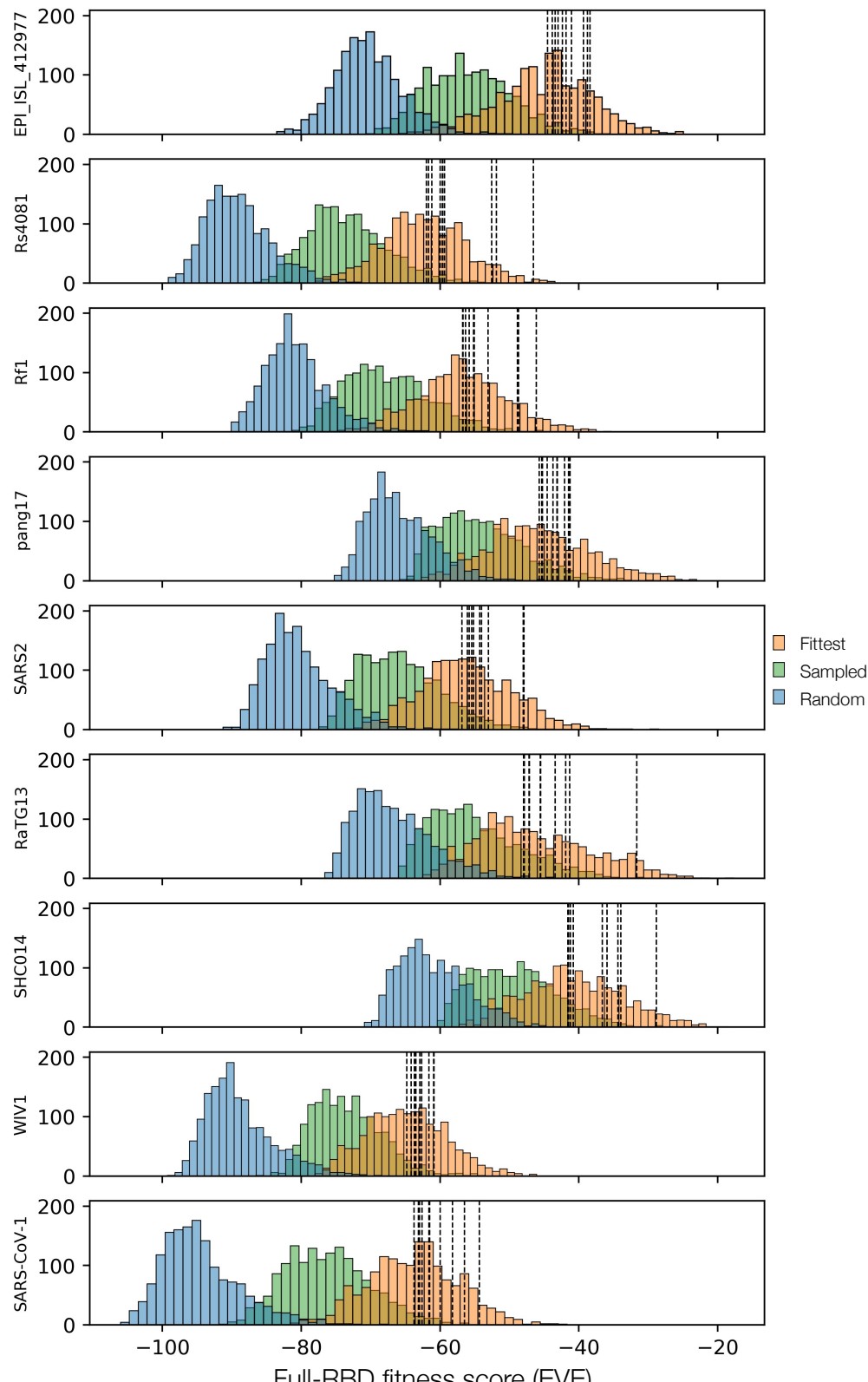

*Figure 5.* For each of the natural RBDs in the mosaic-8b nanoparticle vaccine, we generate 15k sequences (5k by randomly sampling amino acids, 5k by assigning fittest amino acid, 5k by sampling proportional to fitness). Full-RBD fitness scores for 10 selected designs (dashed lines) on each of the nine target sequences sequence, relative to distributions of designs $\mathbb{S}_{\text{fittest}}$, $\mathbb{S}_{\text{sampled}}$, and $\mathbb{S}_{\text{random}}$. Note that some lines are overlapping.

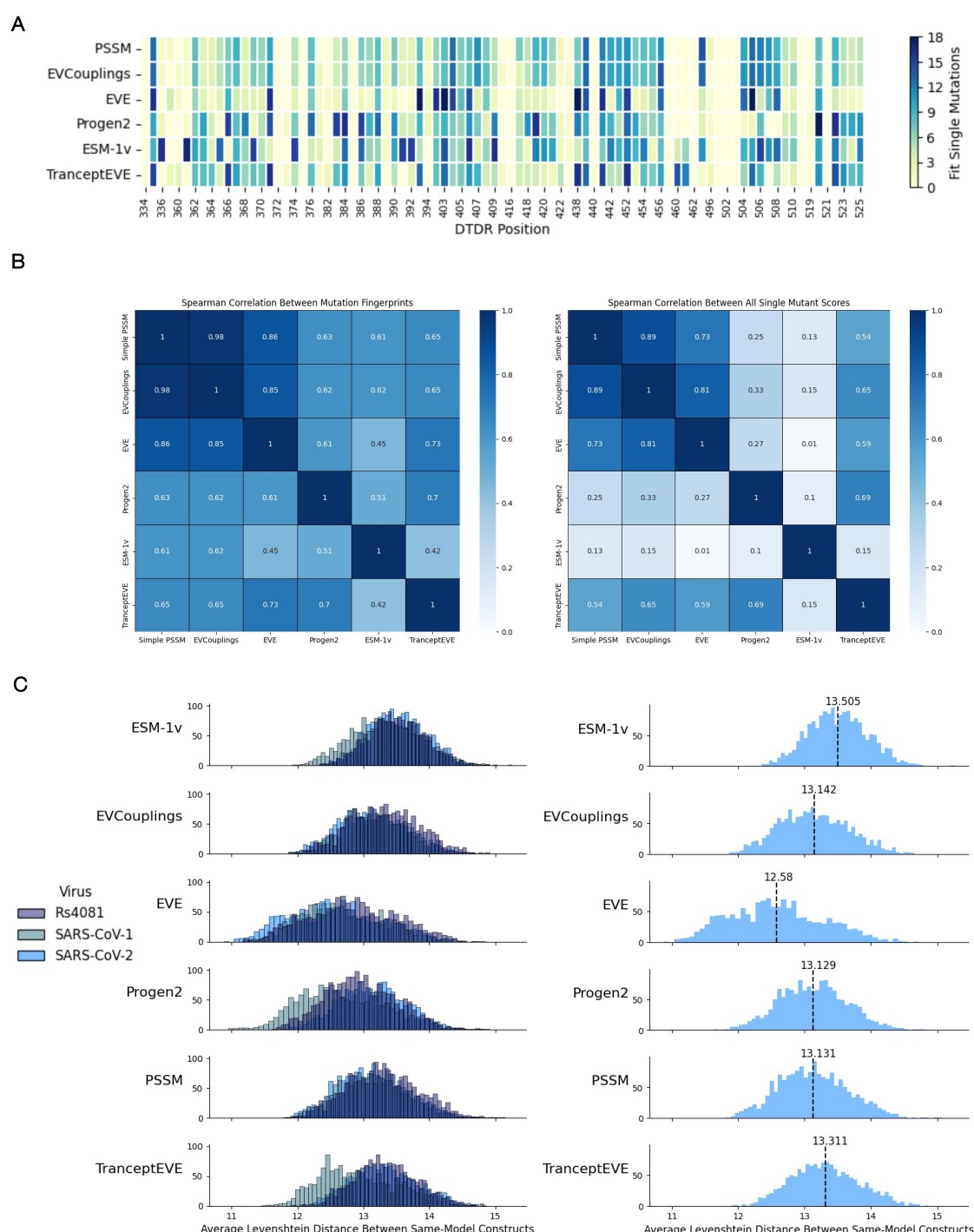

*Figure 6.* **A**. EVE favors specific positions more than other alignment-based sequence fitness models. This heatmap visualizes the number of possible mutations which were observed at each position within the SARS-CoV-2 design set for a given model. The theoretical maximum number of mutations that could be observed at a specific position is 19, since we do not count the wild-type amino acid as a mutation. Position-specific scoring matrix (PSSM), EVCouplings (Hopf et al., 2017), and TranceptEVE (Notin et al., 2022) show relatively weak preference for specific mutational sites, while EVE (Frazer et al., 2021) shows strong preferences for a few mutational sites. Progen2 (Nijkamp et al., 2023) and ESM-1v (Meier et al., 2021) also show moderate preferences for some dTDR sites. **B**. Alignment-based models generate more similar mutational fingerprints than language models. Spearman correlations between mutational fingerprint in set of SARS-CoV-2 designs generated by each model (left) and SARS-CoV-2 single mutation scores (right). **C**. EVE generates less diverse designs than other models. Distributions of average pairwise Levenshtein distances for three target viruses (left) and SARS-CoV-2 only (right). The dotted line shows the mean of all average pairwise Levenshtein distances between designs.

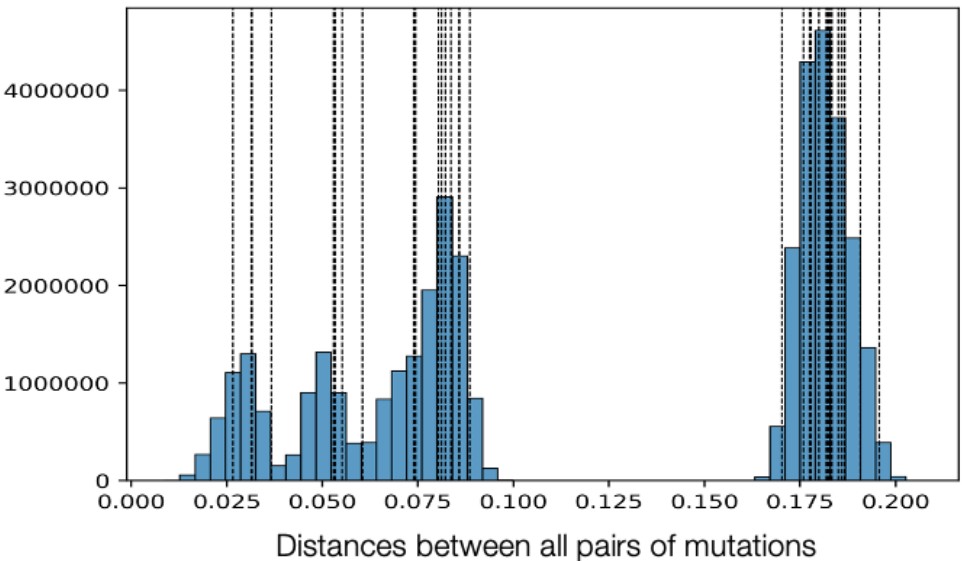

*Figure 7.* Joint accessibility and physicochemical dissimilarity score distances between all pairs of designs in representative nanoparticle (dashed lines) and across all generated designs (blue distribution).

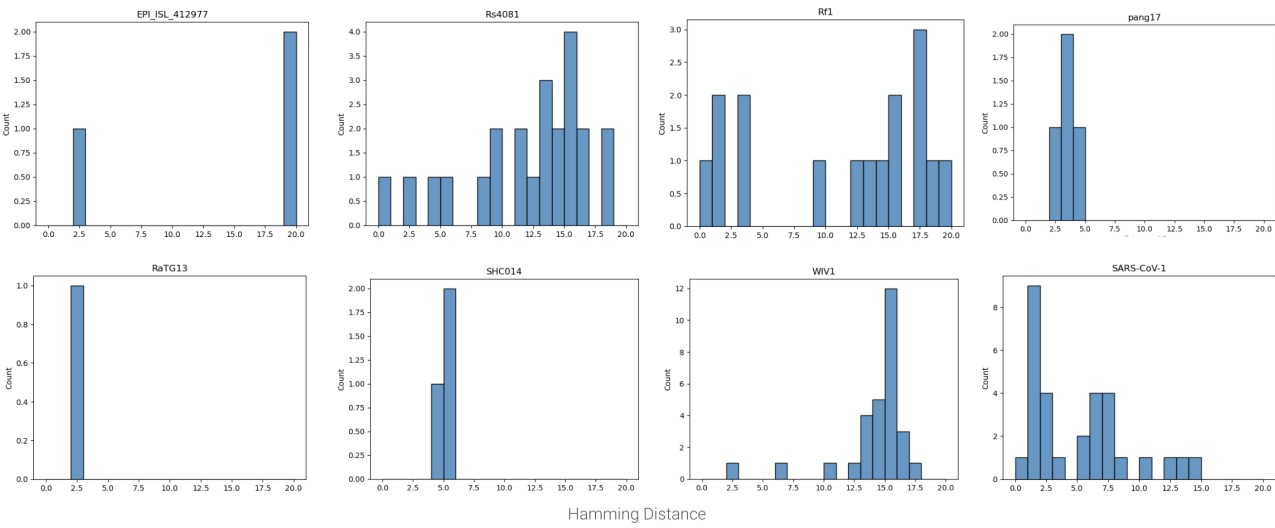

*Figure 8.* Hamming distance between each sequence in aggregated sarbecovirus sequence alignment (within 20 mutations) to each of nanoparticle natural strains. Most species have little local sequence diversity.

# Extended Methods

**Viral Background Sequences**   We chose to design two sets of vaccine designs: one set designing mutations on the 8 sarbecoviruses used by Cohen et al. (2021b) in their original mosaic-8b nanoparticle (SARS-CoV-2 Beta, RaTG13, Rs4081, SHC014, Pang17, RmYN02, Rf1, WIV1), and another set containing designed sequences on SARS-CoV-1 instead of SARS-CoV-2. Altogether, we call these the 9 target viruses. We decided to create groups of designs containing SARS-CoV-1 to reduce the chance of generating recall antibodies in individuals with previous SARS-CoV-2 infection or mRNA vaccination, which may diminish preferred antibody responses (Cohen et al., 2024; Aguilar-Bretones et al., 2023). Note, for the purposes of clarity of the main text we focus there only on the original 8 sequences (containing SARS-CoV-2).

**Protein Structures**   We selected a structure from the Protein Data Bank (7C01), which shows the SARS-CoV-2 Spike RBD in complex with a neutralizing antibody (a relevant orientation for vaccine design), and provides full coverage of the desired RBD residues. This structure is used for all distance and accessibility calculations within the RBD. For calculating the distance of each residue to the ACE2 receptor we selected 6M0J, where chain A is the ACE2 receptor and chain B is the RBD.

**Multiple Sequence Alignments**   We use JackHMMER and super5 to construct multiple sequence alignments (MSAs) for each of our 9 target viruses (Johnson et al., 2010; Edgar, 2021). For each virus, we create a alignment using 5 JackHMMER iterations to search against the Uniref100 dataset (Suzek et al., 2007) with bitscore 0.1 normalized to the full sequence length. To avoid bias due to large numbers of SARS-CoV-2 pandemic sequences in UniRef, we remove all SARS-CoV-2 sequences. We concatenate the 9 alignments to create a combined alignment, so all sequences identified by JackHMMer for each target virus will be considered for all target viruses. We use super5 to realign the combined alignment to each target virus, creating 9 final individual sarbecovirus alignments. Lastly, we filter the individual alignments by removing any sequences with less than 50% match to the target virus and removing any residue positions with more than 70% gaps.

**Sequence Fitness**   To estimate the fitness effect of substitution mutations, and score full-RBD sequences, we train an EVE model (Frazer et al., 2021) on the final sarbecovirus MSA for each target virus. EVE is a Bayesian variational autoencoder (VAE) capable of capturing higher-order interactions across sites that learns constraints underpinning structure and function from evolutionarily-related sequences. The fitness of a protein sequence is quantified using the log likelihood ratio of the mutated sequence over that of the wildtype sequence. EVE is comprised of a symmetric 3-layer encoder and decoder architecture (with 2,000-1,000-300 and 300-1,000-2,000 units respectively) and a latent space of dimension 50.

Thadani et al. (2023) demonstrated that EVE (and EVCouplings) is highly successful at learning constraints important to viral fitness (Starr et al., 2020) and antibody escape (Greaney et al., 2021), especially frequent pandemic mutations (Shu & McCauley, 2017). Moreover, Youssef et al. (2023) showed that EVEscape (and EVE) can be used to design infectious pseudovirus Spike proteins with novel combinations of up to 10 mutations from existing Variant of Concern strains, and up to 46 mutations from the ancestral Wuhan strain.

Notably, even when training on an alignment of natural RBD sequences (not the Spike), the EVE model learns constraints underlying contacts with the full Spike. These constraints are not present in deep mutational scanning experiments of RBD expression alone (Starr et al., 2020). This results in some discrepancies between model and experiment, where the models alone capture structural constraints that are meaningful in real life infections with a full Spike protein (Thadani et al., 2023). However, these are in fact unnecessary constraints in the case of RBD nanoparticle design–especially as we have a goal of mitigating off-target antibody binding in full Spike contacts with the RBD.

To run EVE, we use a theta re-weighting value of 0.01 to cluster similar sequences. We use these 9 EVE models to score all possible single mutations for the RBD of each target virus. Based on these fitness scores, we label each single mutant as above or below a 50th percentile fitness score cutoff. We use a 50% cutoff as as strict cutoff based on previous deep mutational scans of the SARS-CoV-2 RBD expression, which indicate that ∼30% of single mutations result in a non-functional RBD (Starr et al., 2020). We use the fitness scores to select mutation sites with at least one fit mutation, and then to sample fit mutations in our design process that result in functional RBDs with 8 mutations from each target virus. We also score full-RBDs with the EVCouplings fitness model (Hopf et al., 2017) to further validate the fitness of our choices of designed sequences.

We also explore the use of alternate fitness models. For alignment-based models, we consider both EVCouplings (Hopf et al., 2017), which considers pairwise relationships between residues, as well as a simple position-specific scoring matrix (PSSM) which considers each column of the MSA in isolation. We also consider Progen2 (Nijkamp et al., 2023) and ESM-1v (Meier et al., 2021), two state-of-the art

transformer-based protein language models which use natural language processing techniques to predict the likelihood of observing a given protein sequence given the lexicon of existing protein sequences. However, viral sequences are relatively underrepresented in the protein sequence universe, especially as only sequences from Uniref90 (which doesn't consider separately sequences that are more than 90% identical, as is true for much of viral variation) are included in training. Finally, we include TranceptEVE (Notin et al., 2022), a hybrid model which combines language modeling with alignment-based modeling. By repeating our design strategy with multiple diverse models, we gain insights into the quality of the designs generated by each model, and the impact of using each model on designed mutations.

**Antibody Accessibility** The antibody accessibility of each position is computed from its negative weighted residue-contact number from an RBD structure, capturing both protrusion from the core structure and conformational flexibility. Thadani et al. (2023) demonstrated that weighted contact number identifies potential antibody binding sites without prior knowledge of B cell epitopes, and is highly predictive of antibody escape (Greaney et al., 2021) and pandemic mutations (Shu & McCauley, 2017). We use the accessibility score in section to select combinations of designs with cross-reactivity at each off-target epitope optimally blocked–with residues and epitopes weighted by their antibody accessibility.

**Physicochemical Dissimilarity** The dissimilarity between two amino acids is computed using differences in hydrophobicity (Eisenberg et al., 1984) and charge, chosen as physicochemical properties impacting protein-protein interactions. Thadani et al. (2023) demonstrated that this physicochemical score predicts which amino acids within a known escape site actually result in antibody escape (Greaney et al., 2021). We use this physicochemical amino acid similarity matrix to identify a physicochemically conserved region in section and select optimal combinations of designs that eliminate cross-reactivity (as determined by their physicochemical dissimilarity) at off-target epitopes in section .

IDENTIFY TARGET AVIDITY BINDING REGION

We define the Spike receptor binding domain (RBD) as containing residues 334-525 of the SARS-CoV-2 Spike protein, following Cohen et al. (2021a). We divide the RBD into two main regions: the Target Avidity Region (TAR) and the Target Disruption Region (TDR). The TAR contains residues which are likely to elicit neutralizing antibodies, but which are conserved throughout sarbecoviruses (and so are unlikely to mutate rapidly) and are accessible in a full Spike protein during an infection (not in an RBD position

that is in contact with the rest of the Spike protein). The TDR contains all residues outside the TAR. Our goal is to induce point mutations throughout the TDR, disrupting antibody binding and forcing antibodies to bind to the TAR. We anticipate that the immune system will preferentially elicit antibodies that can cross-link two TARs on adjacent, different RBDs on the nanoparticle due to avidity effects. This will result in increased production of broadly neutralizing sarbecovirus antibodies.

To identify TAR residues, we first score all RBD residues based on two metrics: ACE2 binding proximity and physicochemical-based conservation. We take a rank product of ACE2 binding proximity and physicochemical conservation to generate an overall neutralization-conservation (NC) score for each residue, which prioritizes residues that are both highly conserved and proximal to the ACE2 binding site (e.g., binding likely neutralizing antibodies). To generate a continuous surface for antibodies to bind within the TAR, we use a patch-based approach to select residues to include in the TAR. First, we define a patch around each non-Spike contacting RBD residue containing all residues with side chain centers of mass within 8Å. We calculate the average and standard deviation of NC scores for each patch and select all patches within the top 30% of average NC scores and the bottom 30% of standard deviation NC scores across patches. This group of patches constitutes the initial TAR. To improve continuity of potential antibody binding sites, we expand the TAR to include any residues within 5Å of at least two positions in the initial TAR. As a final step to remove gaps, we include any one or two linearly adjacent positions which are linearly surrounded by TAR residues (in the primary sequence). This results in a final TAR of 66 residues, which is about one-third of the entire RBD.

**ACE2 binding proximity** As a proxy for the neutralizing potential of antibodies binding to a given residue, we calculate the distance to ACE2 for each residue, as most RBD neutralizing antibodies inhibit viral binding to ACE2, thus blocking viral entry. For ACE2 binding proximity, we calculate the Euclidean distance of each residue's alpha carbon atom to the center of the ACE2 binding site. We define the ACE2 binding site as the center of mass of all residues which have at least one atom within 3.5Å of at least one atom on the ACE2 receptor.

**Physicochemical-based conservation** We calculate a conservation score for each residue, to identify a conserved region that is therefore less likely to mutate in the future as well as that will permit binding with avidity across target viruses. We found the physicochemical-based conservation score for each residue in the RBD using Von Neumann Entropy on our MSA for each target virus.

Conservation is often measured from multiple sequence

alignments via Shannon Entropy

$$SE = \sum_i f_i \log_{20} f_i$$

using the relative frequency $f$ of each of the 20 amino acids $i$ at a given site in the alignment. This uses logarithm base 20 to bound values between zero and one. The frequencies from the alignment are first weighted (by the inverse of the number of sequences in a sequence cluster) if using a large alignment where there may be sampling bias, as is the case for the sarbecovirus alignments. The frequencies are calculated without including gaps.

However, Shannon Entropy ignores that distinct amino acids can have different levels of similarity, and should not be considered to be completely orthogonal. In order to consider antibody binding similarity we use EVEscape's charge-hydrophobicity score (Thadani et al., 2023) that incorporates physicochemical properties known for impacting protein-protein interactions. The charge-hydrophobicity metric is normalized from 0 to 0.8, with 0 being less similar and 0.8 being more similar, and wildtype residues are given a score of 1.

For Von Neumann Entropy (VNE), the Shannon Entropy equation can be modified by matrix multiplying the diagonal matrix of the frequencies from the alignment with a similarity matrix and doing the entropy calculation instead on the eigenvalues of this matrix, which allows for an importance weighting based on physicochemical similarity (Caffrey et al., 2004). While Caffrey et al. (2004) uses BLOSUM50 (Henikoff & Henikoff, 1992) for a similarity matrix, Thadani et al. (2023) found that the charge-hydrophobicity scores is more predictive of antibody cross-reactivity (Greaney et al., 2021). The final physicochemical-based conservation score at a given site is 1 - VNE, which is then multiplied by the fraction of non-gapped residues at that position.

SELECT MUTATION SITES

To identify Target Disruption Region (TDR) residues wherein mutations will be made, we first consider all RBD residues not in the TAR. From this group of residues, we subdivide the TDR into the Evolutionary Target Disruption Region (eTDR) and the Designed Target Disruption Region (dTDR). The eTDR contains regions of the RBD which already exhibit sufficient evolutionary variation to prevent binding of bivalent antibodies to adjacent RBDs from different viral species. We use the same patch-based approach described above for the TAR to identify the eTDR, but when calculating physiochemical conservation, we consider only variation within the 9 target viruses, rather than broader variation across the sarbecoviruses. Also, we do not consider ACE2 binding proximity when designating the eTDR. We consider all patches in the bottom 10% of physicochemical-based conservation scores, as well as residues within 5 Å

of at least two positions in these patches. This results in an eTDR of 66 residues. The remaining 87 RBD residues constitute the dTDR, the region where we will introduce point mutations to deflect off-target antibody binding. While the region definitions are created using the SARS-CoV-2 RBD numbering, we map each region across to the appropriate residue numbering system of each target virus by using our multiple sequence alignment of target viruses.

After identifying the dTDR residues, our next objective is to select combinations of these residues to mutate. For any given designed RBD, we would like to introduce enough mutations to disrupt antibody binding throughout each potential off-target epitope within the dTDR. To identify the optimum number of mutations to make, we consider the alpha carbon atoms of residues in the dTDR as a set of coordinates in three-dimensional space. We use the SARS-CoV-2 RBD structure as a proxy structure for all target viruses, since there is a high degree of structural similarity between target RBDs across target sarbecoviruses. We perform k-means clustering on the set of dTDR coordinates for different numbers of centroids between 4 and 16, using Elkan's algorithm (Hamerly & Elkan, 2002). We find that using 8 centroids generates clusters in which the maximum distance between residues is no larger than ~25Å, which is the average diameter of conformational B cell epitopes (Sun et al., 2011). Therefore, we infer that 8 mutations is the smallest number of mutations which can effectively disrupt antibody binding across each potential off-target epitope in the dTDR.

In order to optimize diversity, we wish to create many potential RBD designs for each target virus, each with 8 mutational sites per RBD, by using k-means clustering on the set of dTDR alpha carbon Euclidean coordinates with 8 centroids. We run the k-means algorithm with 5000 different random initializations for each target virus to generate sufficient diversity. Within each set of 8 clusters, we select one residue position per cluster as a mutation site. To select a cluster-wise mutation site, we iterate through the cluster's residue positions in order of centroid proximity, selecting the first position where at least one single mutant at this position scores above the 50% EVE fitness threshold for the target virus in question. If no residue positions in a given cluster satisfy this constraint, we do not select any residue position for that cluster. After repeating this process for all centroid initializations, we end up with 5000 designed mutation site sets for each of our 9 target viruses. However, the majority of these sets do not contain 8 distinct mutation sites, since many sets contain at least one low-fitness cluster where no mutation site was selected. After removing all mutation sets which contain fewer than 8 distinct mutation sites, the results in approximately 1700 mutation site sets per target virus.

SAMPLE MUTATIONS

We are now able to generate designed sequences by selecting a single amino acid mutation at each mutational site. We use the EVE single-mutant scores described above in section to guide this process. We generate three versions of each of the ∼1700 designed mutation site sets per target virus. First, we select the fittest possible single mutation at each of the 8 mutation sites. We call the set of all designed sequences for a target virus generated with this method $\mathbb{S}_{\text{fittest}}$. Next, we select a random mutation at each mutation site without considering fitness. We call the set of designs generated with this method $\mathbb{S}_{\text{random}}$. Finally, we construct a vector for each mutational site, consisting of fitness scores for all possible single mutations at that site which are above the 50th percentile cutoff for single-mutant scores across the RBD. We normalize fitness scores between 0 and 1 within each mutational site. We consider the set of normalized scores as a probability vector, which we use to select a mutation via multinomial sampling. Mutations with a higher fitness score are more likely to be selected, but we allow multiple mutation possibilities at each site in order to introduce diversity across the set of generated designs, particularly in cases where multiple mutations have similarly high fitness. We call the set of designs generated with this method $\mathbb{S}_{\text{sampled}}$. Note that we generate $\mathbb{S}_{\text{fittest}}$, $\mathbb{S}_{\text{random}}$, and $\mathbb{S}_{\text{sampled}}$ separately for each target virus, using our 9 individual EVE models trained on the MSA for each target.

We use the $\mathbb{S}_{\text{fittest}}$ and $\mathbb{S}_{\text{random}}$ distributions in order to select designed sequences from $\mathbb{S}_{\text{sampled}}$ that are diverse yet sufficiently fit. First, we determine the full-RBD EVE fitness score for all designs (each with 8 mutations) in $\mathbb{S}_{\text{fittest}}$ for each target virus. We use the 50th percentile of full-RBD $\mathbb{S}_{\text{fittest}}$ scores as a minimum fitness threshold, and we remove all designs in $\mathbb{S}_{\text{sampled}}$ with EVE fitness scores below this threshold. Second, we remove any remaining designs in $\mathbb{S}_{\text{sampled}}$ with full-RBD EVE fitness scores which are less than the highest full-RBD score in $\mathbb{S}_{\text{random}}$. We cross-check this downselection process by scoring each design also with EVcouplings (Hopf et al., 2017). We remove any design which do not meet these fitness benchmarks (fitter than the 50th percentile of $\mathbb{S}_{\text{fittest}}$, and fitter than the highest-scoring sequence in $\mathbb{S}_{\text{random}}$) when scored with both EVE and EVcouplings.

OPTIMIZE COMBINATIONS OF DESIGNS

The final step of designing a nanoparticle vaccine is selecting a group of 8 sequences, one per target virus, for construction of a mosaic-8 nanoparticle. Each mosaic group may contain either SARS-CoV-2 or SARS-CoV-1, but not both, since we would like to perform separate experiments with SARS-CoV-1 nanoparticles to investigate their effect on the production of recall antibodies. When selecting our

final mosaic group, we optimize our design choice in a pairwise manner, as our goal is to minimize off-target cross-reactivity of epitopes in TDR between pairs of sequences on the nanoparticle. This will minimize antibodies that can bind with avidity to the TDR, and so preferentially elicit antibodies to the more conserved TAR.

**Pairwise Distance** In order to calculate pairwise distances between all designed sequences, we first remap designs from all target viruses onto the SARS-CoV-2 RBD sequence so they can be easily compared. We use a patch-wise approach to define the pairwise distance between sequences, as each patch represents a potential off-target cross-reactive epitope. We define a potential epitope patch $\mathbb{E}$ around each TDR residue $R$ as the set of all neighboring sites $r$ within 10.58Å of $R$, by calculating distances between sidechain centers of mass. 10.58Å was selected as the maximum distance between the sidechain centers of any two linearly adjacent residues in the SARS-CoV-2 RBD. After defining potential epitope patches, we can compare the same epitope patch on any pair of designs using an accessibility-dissimilarity distance metric. We use EVEscape's physicochemical dissimilarity score $\boldsymbol{P'}$, which considers differences in charge and hydrophobicity between amino acids, and accessibility score $\boldsymbol{a}$, which uses the weighted contact number of residues to identify surface exposed and protruding positions where antibodies are likely to bind (Thadani et al., 2023). The joint accessibility-dissimilarity distance is calculated as the average of the products of dissimilarity and accessibility across all residues in an epitope:

$$\text{dist}[e, \boldsymbol{s}_j, \boldsymbol{s}_k] = \frac{1}{|e|} \sum_{r \in e} \boldsymbol{a}[r] \cdot \boldsymbol{P'}[\boldsymbol{s}_j[r], \boldsymbol{s}_k[r]]$$

By combining these two scores, we are able to consider the physicochemical dissimilarity between amino acids in the two sequences at a given position, and then weight by the antibody accessibility of that position. We finally aggregate distance scores to get a distance per pair of designs as the weighted average over all epitopes, where the joint accessibility-dissimilarity score for each epitope is weighted by the average accessibility score of that epitope.

$$\text{acc}[e] = \frac{1}{|e|} \sum_{r \in e} \boldsymbol{a}[r]$$

**Mosaic Nanoparticles** To select a mosaic group of 9 sequences, we use the joint accessibility-dissimilarity pairwise distance metric described above to create a distance matrix between all pairs of designs. We then follow a greedy solution to the maximum dispersion problem, which seeks to find a set of 9 sequences from the complete set of designs such that the sum of pairwise distances between the selected designs are maximized. Note that so pairs of designs from

the same target virus will never be chosen, we consider these pairs of designs to have a negative distance value, and hence all 9 sequences will be from a different target virus. For our greedy approach, we start in turn with adding each possible sequence $s$ and adding it to the mosaic nanoparticle $\mathbb{N}$. We then add to $\mathbb{N}$ by iteratively selecting the next design which maximizes the minimum distance to all designs already in $\mathbb{N}$. When selecting SARS-CoV-2 sequences, we do not consider distance to SARS-CoV-1 sequences, and vice versa, since no mosaic group will contain both viruses.

**Experimental Validation**   The above process generates a potential mosaic nanoparticle $\mathbb{N}$ for each possible starting sequence $s$. We repeated this process with each starting sequence and ultimately selected 10 nanoparticle designs for experimental validation, which have the highest rank sum of distances among all chosen nanoparticles. Designs will be characterized first at the individual RBD level for expression and non-aggregation, as well as binding to a class of known broadly neutralizing antibodies. After passing these initial checks, designs will be constructed as a mosaic RBD nanoparticle vaccine, and used to immunize mice. The resulting sera will be characterized for its breadth of neutralization against existing and potential future sarbecoviruses.