# OpenReview forum: "Future-proof vaccine design with a generative model of antibody cross-reactivity"
_ICML.cc/2024/Workshop/ML4LMS — ML4LMS Poster_

### Official Review · Reviewer_vdrq · 2024-06-08
**Novel idea, but premature for publication**

**Rating:** 4
**Confidence:** 4

**Review:**

The design process for future-proof vaccine generation seems promising, however, the evaluation seems premature for publication. It is unclear whether the chosen metric (nanoparticle expression) correlates with the paper's goal ("a promising approach for pan-coronavirus vaccines"). While it is understandable that mice immunization is a long and expensive process, as a reader I would expect more evidence that the sampled RBDs indeed show potential of being strong vaccine candidates. It was also unclear to me how were the different language models fine tuned (if at all) and on what task they were evaluated on. While the selection and mutation process is clear, the suggested model architecture lacks detail.

---

### Official Review · Reviewer_j7oZ · 2024-06-09

**Rating:** 9
**Confidence:** 4

**Review:**

The authors provide a compelling workflow for applying variant effect prediction models to the development of novel broadly neutralising antibodies via the design of mosaic nanoparticle vaccines.

The manuscript is well-written, clear, and presents a novel application.

*Pros:*
- Though this is a work in progress, the manuscript clearly indicates what has been completed and what remains to be completed, with genuine insights from the work completed thus far.
- The work proposes a useful practical application for well-known variant effect prediction models.
- The work has proposed a robust in vitro evaluation framework and offered thoughtful interim analyses.
- The work takes a model-agnostic approach and provides evaluations of alternative models.

*Cons:*
- (minor) The authors are somewhat overloading the term "off-target". In this manuscript it refers to an undesirable epitope on an intended antigen, but more commonly in the literature it refers to unintended antigens with connotations of toxicity. Suggest using an alternate phrasing.
- (minor) The method appears to rely on epitopes being contiguous in sequence. This is eminently reasonable, but it could be mentioned that there may be instances where desirable conformational epitopes are surface-contiguous but not sequence-contiguous and a more complicated identification is required.
- (minor) The definition of "neutralisation potential" here is subjective and specific to sarbecoviruses. Some mention of how this could be generalised would be nice.
- (minor) Typos in Section 3. P1.

---

### Official Review · Reviewer_5epy · 2024-06-10
**Interesting work on exploring RBD mutations**

**Rating:** 5
**Confidence:** 3

**Review:**

This paper introduces a method for exploring epitope mutations and leveraging these RBD designs for better vaccine development.
The joint accessibility and physicochemical dissimilarity metric is quite interesting and I could imagine it finding useful applications beyond the scope of this work.
The work presented is interesting and quite creative but is of limited technical novelty and the applicability of the method presented in real use cases is not completely obvious from the current results and the paper is lacking some form of validation beyond an ML fitness score. The authors mentioned that experimental validation is on-going, but lacking those some further in silico studies would be useful, e.g. evaluating the impact of identified mutations on binding using physics based methods.
Minor comments:
- in section 2.3, the three sampling strategies are listed in the wrong order with respect to how they are then described
- in section 2.3, the authors describe using a mutational score, but it's not clear to me how this score is obtained, is this an output of EVE or of the clustering, and how is it computed?
- how is the norm of an epitope |e| defined, is this the length of the sequence?
- below table 1, sampling is spelt incorrectly